# Denisovan introgression has shaped the immune system of present-day Papuans

**Davide M. Vespasiani**[1,2], **Guy S. Jacobs**[3], **Laura E. Cook**[1,2], **Nicolas Brucato**[4], **Matthew Leavesley**[5,6,7], **Christopher Kinipi**[5], **François-Xavier Ricaut**[4], **Murray P. Cox**[8], **Irene Gallego Romero**[1,2,9,10] *

1 Melbourne Integrative Genomics, University of Melbourne, Parkville, Australia, 2 School of Biosciences, University of Melbourne, Parkville, Australia, 3 Department of Archaeology, University of Cambridge, Cambridge, Uniteed Kingdom, 4 Laboratoire de Evolution et Diversite Biologique, Université de Toulouse Midi-Pyrénées, Toulouse, France, 5 School of Humanities and Social Sciences, University of Papua New Guinea, Port Moresby, Papua New Guinea, 6 College of Arts, Society and Education, James Cook University, Cairns, Australia, 7 ARC Centre of Excellence for Australian Biodiversity and Heritage, University of Wollongong, Wollongong, Australia, 8 School of Natural Sciences, Massey University, Palmerston North, New Zealand, 9 Center for Stem Cell Systems, University of Melbourne, Parkville, Australia, 10 Center for Genomics, Evolution and Medicine, University of Tartu, Tartu, Estonia

* irene.gallego@unimelb.edu.au

**Data Availability Statement:** All requests to access the sequences from Jacobs et al are managed through the Data Access Committee of the official data repository (accession

## Abstract

Modern humans have admixed with multiple archaic hominins. Papuans, in particular, owe up to 5% of their genome to Denisovans, a sister group to Neanderthals whose remains have only been identified in Siberia and Tibet. Unfortunately, the biological and evolutionary significance of these introgression events remain poorly understood. Here we investigate the function of both Denisovan and Neanderthal alleles characterised within a set of 56 genomes from Papuan individuals. By comparing the distribution of archaic and non-archaic variants we assess the consequences of archaic admixture across a multitude of different cell types and functional elements. We observe an enrichment of archaic alleles within cis-regulatory elements and transcribed regions of the genome, with Denisovan variants strongly affecting elements active within immune-related cells. We identify 16,048 and 10,032 high-confidence Denisovan and Neanderthal variants that fall within annotated cis-regulatory elements and with the potential to alter the affinity of multiple transcription factors to their cognate DNA motifs, highlighting a likely mechanism by which introgressed DNA can impact phenotypes. Lastly, we experimentally validate these predictions by testing the regulatory potential of five Denisovan variants segregating within Papuan individuals, and find that two are associated with a significant reduction of transcriptional activity in plasmid reporter assays. Together, these data provide support for a widespread contribution of archaic DNA in shaping the present levels of modern human genetic diversity, with different archaic ancestries potentially affecting multiple phenotypic traits within non-Africans.

EGAS00001003054) at the European Genome-phenome Archive (EGA; https://www.ebi.ac.uk/ega/home). All remaining data are within the manuscript and its Supporting information files. All analyses were performed using R software v 4.0.0. The complete list of scripts used for the analyses is available at https://gitlab.unimelb.edu.au/igr-lab/Archaic_introgression_in_PNG.

**Funding:** This work was supported by an award from the Leakey Foundation and by Australian Research Council Discovery Project DP200101552, both to I.G.R. and by The French National Research Agency (ANR) (grant PAPUAEVOL n° ANR-20-CE12-0003-01 (F.X.R). N.B. was supported by the PAPUAEVOL grant. D.M.V was supported by the University of Melbourne's Albert Shimmins Fund. The funders had no role in study design, data collection and analysis, decision to publish, or preparation of the manuscript.

**Competing interests:** The authors have declared that no competing interests exist.

## Author summary

Humans of Papuan ancestry owe roughly 5% of their genome to Denisovans, a poorly characterised archaic hominin. While introgressed DNA segments can be readily identified, understanding their biological consequences remains challenging. By examining the distribution of introgressed DNA against existing functional genomics datasets, it is possible to predict the phenotypes they impact. In Papuans, Denisovan DNA, but not Neanderthal, strongly and consistently affects immune cells and immune-related processes of potential evolutionary relevance. In vitro testing of introgressed variants confirms these predictions, suggesting Denisovan variants can impact gene regulation in vivo. Variation in gene expression might be key to understanding the consequences of admixture between modern humans and archaic hominins, as has been observed with Neanderthal DNA in other human populations.

## Introduction

Modern humans are known to have interbred with Neanderthals [1], Denisovans [2] and possibly other archaic hominins [3]. While genetically similar populations of Neanderthals are thought to have contributed approximately 2% to non-African genomes, Denisovan introgression has been observed to be more variable [4]. Particularly, Denisovan ancestry accounts for up to 5% of the genomes of the Indigenous peoples of Island Southeast Asia and Australia [4, 5]. In addition, these components exhibit a deep divergence from the reference Altai Denisovan genome, providing strong evidence for the occurrence of multiple Denisovan introgression events across time and space [6, 7].

At the genomic level these introgressed archaic alleles are mostly observed outside protein-coding sequences, distributed over non-functional and regulatory regions [8, 9]. Enhancers, in particular, are amongst the top targeted elements of archaic hominin introgression [10, 11]. Here, archaic alleles are thought to drive phenotypic differences by altering gene pre- and post-transcriptional regulatory processes [11]. Furthermore, both Neanderthal and Denisovan variants seem to preferentially affect enhancers in a tissue-specific manner, with highly pleiotropic elements being depleted of archaic variation [12]. However, beyond general agreement that introgressed archaic DNA has mainly been deleterious and actively removed from coding sequences and conserved non-coding elements [8, 12–14], the actual phenotypic consequences of this variation are not well understood. Several lines of evidence highlight associations between archaic DNA and risk for disease traits, including autoimmune diseases [8, 15, 16], or with traits of possible evolutionary advantage for early non-Africans [17–20]. For example, Neanderthal variants within immune genes and immune-related cis-regulatory elements (CREs) have been associated with differential responses to viral infections among present-day Europeans [21, 22].

Unfortunately, two main factors limit our interpretation of the biology of these alleles. First, we still lack a detailed characterisation of global levels of human genetic diversity, vital to identifying differential archaic hominin contributions across modern populations [23, 24]. Because Denisovan DNA is not present in the genomes of modern Europeans, which make up the vast diversity of biomedical genetics cohorts, it has been impossible to link them to phenotype, unlike segregating Neanderthal variants [16, 25]. Second, and more challenging, most of these alleles lie within non-coding sequences where, despite their acknowledged contributions to human evolutionary history [26, 27], an understanding of their actual biological functions remains elusive.

To gain insights into the consequences of archaic introgression in Papuans, we have analysed a previously published dataset of 56 present-day genomes sampled across the island of New Guinea [6]. By comparing the distribution of archaic single nucleotide polymorphisms (aSNPs) and non-archaic SNPs (naSNPs) segregating within these populations, across multiple genomic elements and cell types, we find that aSNPs are enriched within functional cis-regulatory elements (CREs) and transcribed regions, particularly those active within immune-related cells. We also observe that the presence of archaic alleles within these elements can lead to substantial disruption of transcription factor binding sites (TFBSs), with a consistent Denisovan-specific signal affecting immune-related processes. We finally validate these results through experimental testing of multiple Denisovan aSNPs, and find they are associated with significant transcriptional changes in reporter gene experiments.

## Materials and methods

### Ethics statement

All human genome data in this study was taken from Jacobs *et al.* [6]. All collections followed protocols for the protection of human subjects established by institutional review boards at the Eijkman Institute (EIREC #90) and Nanyang Technological University (IRB-2014–12-011); the analyses in this publication were additionally approved by University of Melbourne's Human Ethics Advisory Group (1851585.1). Permission to conduct research in Indonesia was granted by the State Ministry of Research and Technology (RISTEK). All individuals gave their full informed written consent to participate in the study.

The two lymphoblastoid cell lines (LCLs) used in the plasmid reporter experiments were established from donors sampled at the University of Papua New Guinea, in Port Moresby, Papua New Guinea. Collection of material from healthy adult donors was coordinated by Dr Christopher Kinipi (Director of Health Services at the University of Papua New Guinea) and approved by the Medical Research Advisory Committee of the National Department of Health of the Government of Papua New Guinea (permit number MRAC 16.21), by the University of Melbourne (ID: 1851585.1), and by the French Ethics Committees (Committees of Protection of Persons 25/21_3, n˚SI:21.01.21.42754). Permission to conduct research in Papua New Guinea was granted by the National Research Institute of Papua New Guinea (permit 99902292358), with full support from the School of Humanities and Social Sciences, University of Papua New Guinea. These approvals commit our team to following all ethical guidelines mandated by the government of Papua New Guinea. All individuals gave their full informed written consent to participate in the study.

### Variant filtering and selection

A list of 7,791,042 SNPs segregating within non-archaic, Denisovan and Neanderthal haplotypes identified in 56 genetically Papuan individuals was taken from [6] (for detailed information about samples, genotyping and inference of introgressed state, see [6]). To define a high confidence set of introgressed SNPs and a matched set of SNPs found outside archaic haplotypes to use as controls, we applied a series of stringent filtering steps to the genome-wide data, as follows. First, all singletons were removed, together with variants lacking ancestral state information in the 6 non-human primate EPO alignment [28]. Variants were then labelled as possibly archaic if they fell within archaic haplotypes defined by [6], and further classified as Denisovan or Neanderthal aSNPs on the basis of the inferred ancestry of the haplotype. Due to incomplete lineage sorting and the relatively short divergence time between Denisovan and Neanderthal lineages [2], a single archaic SNP can segregate in both Neanderthal and Denisovan haplotypes. We observed 18,518 aSNPs that segregated in both Denisovan and

Neanderthal haplotypes. These variants were assigned to one of the two archaic ancestries first by considering in which archaic haplotype they segregated at higher frequencies and, then, by comparing the state of the main introgressed allele in Papuans to the Altai Neanderthal [29] or the Altai Denisovan [2] reference genomes. This resulted in 6,098 and 9,744 variants that were respectively assigned to Denisovan and Neanderthal ancestries. The 2,676 variants that could not be disambiguated in this manner were labelled as ambiguous and excluded from downstream analyses.

To further control for incomplete lineage sorting between humans, Neanderthal and Denisovans, and possible errors in haplotype calling, allele frequencies were calculated by considering the number of archaic or non-archaic haplotypes carrying the allele of interest relative to the total number of chromosomes in the sample. Specifically, for naSNPs we calculated the derived allele frequency (DAF) as the number of derived alleles (relative to the EPO alignment) segregating within non-archaic haplotypes divided by the total number of chromosomes genotyped at that site. Likewise, for each aSNP, we calculated the main introgressed allele frequency (MIAF) by counting the number of observations of the most abundant allele found within archaic haplotypes divided by the total number of chromosomes genotyped at that site. Variants with MIAF or DAF $< 0.05$ or $\geq 0.05$ were respectively labelled as low-frequency or common to high-frequency. We also calculated the frequency of the main introgressed allele within non-archaic haplotypes for all aSNPs, and then examined the differences in frequency across modern and archaic haplotype backgrounds. This showed that in a large number of cases, alleles that segregate within archaic haplotypes also segregate within individuals that carry a non-archaic haplotype at the same site. To address this confounder, we calculate allele frequencies in archaic and non-archaic haplotypes, and removed all SNPs where the difference in frequency between archaic and non-archaic haplotypes was $< 0.25$ (S1 Fig). This step does also exclude some aSNPs segregating at overall low frequencies in our samples and thus reduces the overall size of our aSNP sets, but leads to a higher-confidence dataset. Finally, we removed all aSNPs and naSNPs for which the main introgressed allele or derived allele, respectively, segregated at frequencies $\geq 0.005$ within the sub-Saharan African populations in the 1000 Genomes Project [30].

Following these steps, we retained a set of 140,916, 88,625 and 1,108,747 variants of putative Denisovan, Neanderthal and non-archaic ancestry, respectively. A detailed overview of the number of variants retained after each filtering step can be found at S1 Table. Inspection of the site frequency spectrum of these three sets (SFS) revealed a different distribution of the allele frequencies between aSNPs and naSNPs (S2(A) and S2(B) Fig). To account for this, we randomly sub-sampled a total of 229,541 naSNPs (corresponding to the combined number of aSNPs) from the set of non-archaic variants to match the SFS of both Denisovan and Neanderthal aSNPs and used this set for all downstream comparisons (S2(C) Fig). The final sets of Denisovan, Neanderthal and non-archaic variants are listed in S2–S4 Tables.

## Definition of a control set of Neanderthal variants segregating within West Eurasians

We additionally defined a control set of Neanderthal variants following the same principles as above. Briefly, Neanderthal haplotype inferences were taken from 75 West Eurasian individuals from the Simons Genome Diversity Project [31] that were analysed in [6], retrieving the Neanderthal haplotypes inferred and extracting allele counts for each SNP in Neanderthal and non-Neanderthal inferred haplotypes. Filtering steps to obtain a high-confidence set of Neanderthal variants segregating in West Eurasians were performed as above.

## Quantifying the levels of background selection

To ensure the observed differences between aSNPs and naSNPs were not caused by different levels of background selection we downloaded from https://github.com/gmcvicker/bkgd the bed file containing the list of pre-computed values for the B-statistic in hg19 coordinates [32]. We then intersected these regions with the list of aSNPs and naSNPs to test whether the distribution of the corresponding values were comparable across the three ancestries. As a further control, we also calculated the distribution of the B-values for all know human protein-coding sequences (S3(A) and S3(B) Fig).

## Functional annotation

To functionally annotate the refined set of SNPs we first used the annotatr R package [33] to determine the distribution of aSNPs across multiple genomic elements and whether they were significantly depleted from protein-coding sequences. To investigate the function of aSNPs across chromatin states and human cell types, we downloaded the 15-state hg19 mnemonics BED files containing the segmented chromatin state predictions for each of the 111 cell types included in the Roadmap Epigenomics project [34], https://egg2.wustl.edu/roadmap/web_portal/index.html. Variants were intersected with chromatin state files, and the 111 cell types were grouped into the the 18 tissues as in [34]. We counted the number of cell types across which each SNP was annotated within the same chromatin state as a measure of its potential pleiotropic activity. Finally, we downloaded the list of V8 cis-eQTLs and their eGenes from the GTEx web portal (https://www.gtexportal.org/home/) [35], retaining only the positional information for the list of significant cis-eQTLs (i.e., qval < 0.05). Variants associated with multiple genes were counted only once. The genomic coordinates for the resulting list of cis-eQTLs were then lifted over from hg38 to hg19 using liftOver from the rtracklayer R package [36]. We used this dataset to identify shared instances first only with our set of common-to-high-frequency variants annotated within putative CREs and then with our entire set of 140,916 Denisovan, 88,625 Neanderthal and 229,541 non-archaic variants. In both analyses, we retained only those variants where the reference and alternative alleles matched between the datasets.

## Computing $F_{st}$ values between western Indonesia and New Guinea populations

$F_{st}$ values were calculated between western Indonesian populations and New Guinean populations using the BCF files from [6] and the VCFtools –*weir-fst-pop* command. The western Indonesian population set was defined as in [6].

## Enrichment of aSNPs across chromatin states and tissues

To quantify the impact of aSNPs within each chromatin state and/or across each chromatin state/tissue combination, we counted the number of Denisovan, Neanderthal and non-archaic variants annotated in each combination of interest. We then computed the odds ratio (OR) between the number of aSNPs and naSNPs for Denisovan and Neanderthal variants. Statistically significant deviations from an OR = 1 were tested using Fisher's exact test. Unless otherwise noted, resulting p-values were corrected for multiple hypothesis testing using the Benjamini-Hochberg (BH) method [37]. Enrichment scores together with their corresponding p-values for each comparison of interest can be found at S5–S7 Tables.

## Impact of aSNPs on transcription factor binding sites

To quantify the impact of archaic alleles on transcription factor binding sites (TFBSs), we focused on variants annotated to TssA, TssAFlnk, TxFlnk, Enh and EnhG in at least one cell type. We then used the motifbreakR R package [38] to identify putative TFBS-disrupting variants. Position weight matrices (PWMs) were retrieved from Jaspar 2018 [39] and HOCOMOCO v.11 [40]. Motifs were matched to the hg19 genome using p-value threshold of $1^{-5}$, and the allelic impact on each PWM (PWM score) was quantified via the built-in sum of log-probabilities algorithm specifying a background probability of 0.3 and 0.2 for A/T and G/C, respectively. Differences in PWM score between the two alleles at a site, defined as $\Delta$ PWM, were polarised by the introgressed or derived state of aSNPs and naSNPs, respectively. Differences in PWM score between the two alleles at a site, defined as $\Delta$ PWM, were polarised by the introgressed or derived state of aSNPs and naSNPs, respectively. When variants were predicted to disrupt a motif associated with the same TF across both databases we retained only the SNP-PWM association with the highest absolute $\Delta$ PWM score.

To assess the genome-wide impact of aSNPs on TFBSs while avoiding redundancy between TFs due to similar binding motif preferences, we grouped all TFs into families based on motif clustering information from [41]. We removed 67 HOCOMOCO and 62 Jaspar PWMs that were either not associated with any cluster or that have mismatches in the PWM versions between MotifDb [42] and [41]. We then tested whether any motif cluster contained an excess of TFBS-disrupting aSNPs compared to naSNPs by performing a $\chi^2$-test. Differences in $\Delta$ PWM score distributions across ancestries were tested via a Wilcoxon test.

## Gene ontology enrichment analysis

To understand the biological processes potentially affected by Denisovan and Neanderthal introgression, we retrieved all CRE-associated variants annotated in at least one haematopoetic stem cell (HSC) & B and/or blood & T related cell type. We then used the rGREAT R package package [43] to assign each variant to the regulatory domain of the nearest gene located within 1 Mb of distance and to perform a gene ontology (GO) enrichment analysis [44], using as a background set the union of all the TFBS-disrupting variants associated genes (both archaic and non-archaic). Importantly, by linking variants to the regulatory domain of its nearest gene, this approach does not yield multiple SNP-genes associations for a single SNP which could occur when in presence of gene clusters. Significantly enriched GO terms were defined as those with BH-corrected p-value < 0.01 and can be found in S8 Table.

## Quantifying the impact of Denisovan alleles on gene expression using a plasmid reporter assay

For each of the 5 selected SNPs (see Results section), we designed 2 DNA oligonucleotides of 170 nucleotides (nt) centred on the two alleles of interest (84 bp on the 5' and 85 bp on the 3' end). To ensure the effectiveness and reproducibility of our design, we also included a positive control variant (rs9283753) from Tewhey *et al*. 2016 [45]. All variants were synthesised on the positive strand, regardless of variant orientation with respect to its predicted target gene, as enhancer activity in reporter assays has been found to be largely independent of orientation [46]. In addition, all oligos contained two common 15 bp adapter sequences flanking each regulatory element, a 44 bp sequence centred on a BsiWI restriction site, a oligo-specific 15 bp barcode sequence and a universal 20 bp buffer sequence, resulting in a final oligo length of 284 bp. Oligos were synthesised as a pool by IDT. A list with the full sequences and DNA primers

used to amplify the oligo library (see below) is available at S9 Table. The pMPRA1 (Addgene, 49349) was chosen as plasmid backbone for the molecular cloning.

Synthesised oligos underwent two consecutive runs of Q5 (NEB, M0491) low-cycle PCR using two different sets of primers, first to make them double-stranded and then to add a SfiI restriction site at each end of the sequence. After each PCR amplification, oligos were purified with AMPure XP beads (Beckman Coulter, A63881) following the manufacturer's protocol. Amplified oligos and pMPRA1 vector were both SfiI digested (NEB, R0123) for an hour. Digestion solutions were then ran on an agarose gel, extracted using the gel extraction kit (NEB, T1020) and ligated using T4 DNA Ligase (NEB, M0202), again following the corresponding manufacturer's protocol. Ligated products were then transformed into *E.coli* competent cells (Promega, JM109), which were grown overnight in 25 mL of LB media supplemented with 100 $\mu$g/mL of ampicillin (Thermo Fisher Scientific, FSBBP1760–5) prior to plasmid purification (Zymo, D4209). To create the final library, assembled plasmids were linearised using the BsiWI-HF (NEB, R3553) restriction enzyme. An amplicon containing a minimal promoter, GFP open reading frame (ORF), synthesised as a gblock by IDT (S9 Table), was then inserted by Gibson assembly (NEB, E2621). The resulting ligated products were again transformed into bacterial cells, which were overnight grown before subsequent plasmid extraction (Zymo, D4200). For each step, the transformation efficiency was estimated to be $> 10^6$ cfu. We ensured correct assembly of each of the DNA oligos by Sanger sequencing (Macrogen).

The two LCLs used for experimental validation (PNG15 and PNG21) were grown in RPMI (Lonza, 12–702Q) supplemented with 1% Glutamax, 1% NEAA and 10% FBS. Cells were maintained at a cell density of $5 \cdot 10^5$ cells/mL. Each cell line was electroporated in triplicate using the Neon transfection system (Thermo Fisher Scientific, MPK5000). For each transfection, $1 \cdot 10^6$ cells were centrifuged at 700x g and resuspended in 100 $\mu$l of R buffer with 1 $\mu$g of plasmid library before applying 3 pulses of 1200 V for 20 ms each [45]. After transfection, cells were allowed to recover in 2 mL of culture media for 24 hours then collected by centrifugation and frozen at -80˚C. For each transfection replicate, total DNA, including plasmid DNA (pDNA), and RNA were extracted from cells using Qiagen AllPrep kit (Qiagen, 80204) following the manufacturer's protocol, including on-column DNase digestion of the RNA fraction. pDNA and total RNA samples were quantified using the Qbit broad range dsDNA (Thermo Fisher Scientific, Q32850) and RNA kits (Thermo Fisher Scientific, Q10211), respectively. First-strand cDNA was synthesised from 1 $\mu$g of mRNA for each sample using the High-Capacity cDNA Reverse Transcription Kit (Applied Biosystems, 4368814) and an universal custom reverse primers binding downstream to the oligo-specific barcode sequence (see S9 Table).

The regulatory activity of each oligo was then quantified via qPCR using the NEB Luna Universal qPCR Master Mix (NEB,M3003). We first assessed relative primer binding efficiencies through a series of 4 (1:10) serial dilutions of an aliquot of cDNA as in [47]. These efficiencies were then incorporated in the calculation of each oligo's relative expression (see below). For each transfection replicate we quantified the relative abundance of pDNA and cDNA for each oligo using a common forward primer binding within the GFP ORF and 12 different reverse primers, each targeting the oligo-specific barcode sequence. All reactions were performed in triplicate. Relative expression of each oligo for each transfection replicate was calculated with the $\Delta C_t$ method, normalising it by the relative amount of pDNA. Expression differences between archaic and non-archaic alleles were then computed using the relative expression ratio, using a t-test to quantify any significant deviation from an expected ratio of 1 [48], BH-correcting the resulting p-values.

## Results

### Building a high-confidence set of archaic variants

To curate a list of high-confidence archaic SNPs to analyse for possible functional contributions to present- day Papuans, we took advantage of a recently described dataset of both archaic and non-archaic haplotypes segregating within 56 individuals of Papuan ancestry [6]. Due to incomplete lineage sorting and recombination, a substantial fraction of variants segregating in these archaic haplotypes is not expected to be of archaic origin [9]. Thus, to generate a high-confidence set of SNPs of archaic origin (aSNPs) we applied a series of filtering steps to our starting list of 7,791,042 genomic SNPs (see Methods and S1 Table). This allowed us to generate two high-confidence sets of 140,916 Denisovan and 88,625 Neanderthal aSNPs (S2 and S3 Tables, respectively). In parallel we defined a set of 229,541 (the sum of all high-confidence aSNPs) variants segregating in present-day Papuans outside archaic haplotypes and which likely arose after the Out of Africa event that matched the allele frequency distribution of our list of Denisovan and Neanderthal aSNPs (see Methods and S2(C) Fig) to act as a control set in all analyses. We refer to these variants as non-archaic SNPs (naSNPs, S4 Table).

Following calculation of the main introgressed allele frequency (MIAF) and of the derived allele frequency (DAF) for aSNPs and naSNPs respectively (see Methods), we found that 82,013 Denisova, 54,415 Neanderthal and 136,428 non-archaic variants are segregating at common-to-high-frequencies in PNG (MIAF or DAF $> 0.05$, see S1 Table). Importantly, we found that Denisovan and Neanderthal and non-archaic variants, whether segregating at low or common-to-high-frequencies, occur within genomic regions subjected to similar levels of background selection which are also higher (corresponding to less conserved regions) than those observed for protein-coding sequences (S3(A) and S3(B) Fig). We thus reasoned that, despite retaining only 10.5%, 9.2% and 4% of the original set of Denisovan, Neanderthal and non-archaic variants (S1 Table), our filtering criteria yielded a high-confidence set of archaic variants and an adequate corresponding background set of non-archaic alleles. Throughout the text and unless otherwise noted, to prioritise variants of potential evolutionary interest we focused on the above defined set of 82,013 Denisova, 54,415 Neanderthal and 136,428 non-archaic variants common-to-high-frequency variants.

### Similar patterns of introgression for Denisovan and Neanderthal aSNPs across chromatin states

To gain insights into the potential consequences of aSNPs in present-day Papuans, we began by annotating the set of aSNPs and naSNPs across multiple genomic elements (see Methods). We found that SNPs lie within non-coding elements, regardless whether considering the entire set of variants or only those instances segregating at MIAF or DAF $\geq 0.05$ in New Guinea Island (S4(A) and S4(B) Fig). As previously reported [8, 9, 14], we also noted a significant depletion of both Denisovan and Neanderthal aSNPs from protein coding sequences, relatively to naSNPs ($n = 3,431$ and $1,919$, $p = 1.41 \cdot 10^{-11}$ and $6.61 \cdot 10^{-19}$ for Denisovan and Neanderthal aSNPs, respectively).

To investigate the patterns of archaic introgression at a higher granularity, we intersected the set of common-to-high-frequency variants against epigenomic maps of 15 different chromatin states across 111 cell types generated by the Roadmap Epigenomics Project [34], and computed odds ratios (OR) between aSNPs and naSNPs to identify states containing an excess of Denisovan and/or Neanderthal variants. aSNPs are generally depleted from inactive chromatin states (states not associated with gene expression as defined by [34]) as well as promoters and transcription start sites, but enriched within transcribed and subsets of enhancer elements

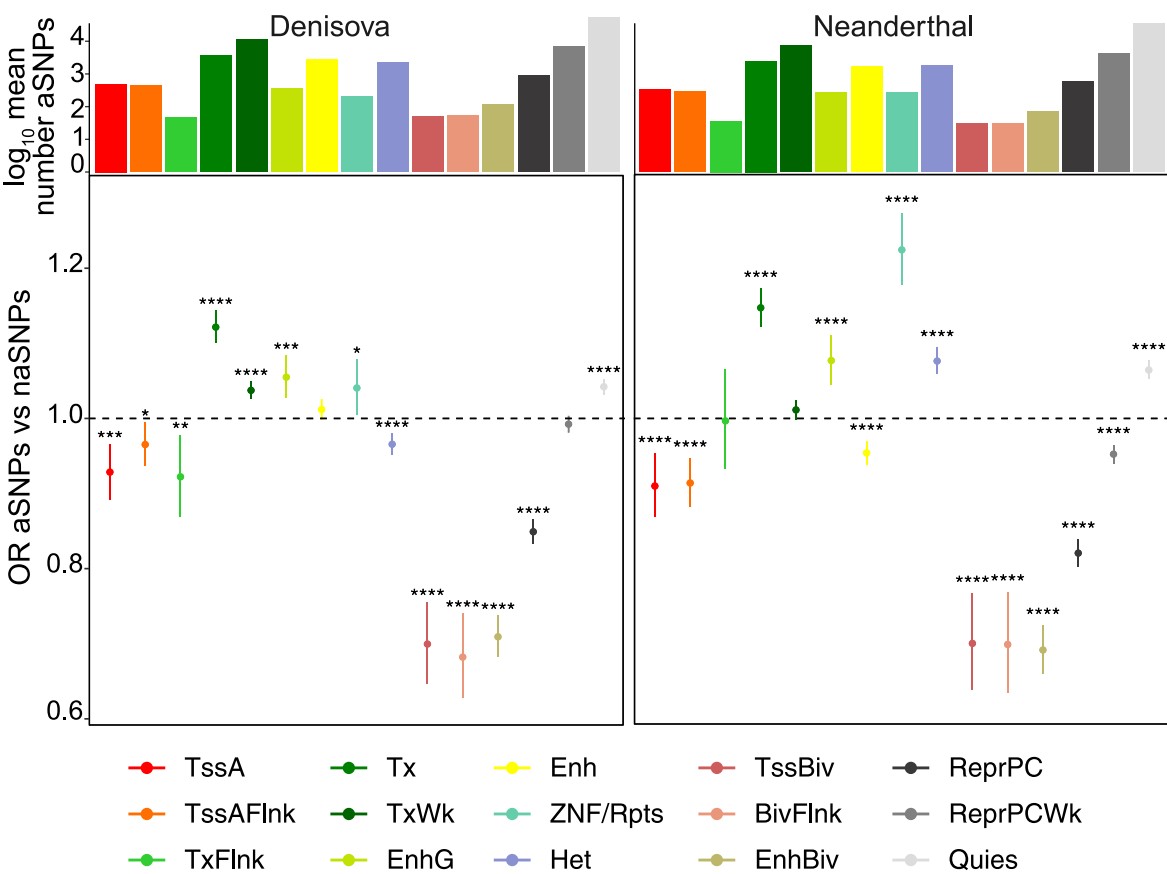

**Fig 1. Distribution of common-to-high-frequency aSNPs across 15 chromatin states and 111 cell types.** Patterns of enrichment across the 15 chromatin states for the set of common-to-high-frequency Denisovan and Neanderthal aSNPs relatively to the matched background set of naSNPs. Histograms on top indicate the mean number of variants within each chromatin state across all 111 cell types. Asterisks indicate BH-corrected Fisher's exact test p-values: $^* = p \leq 0.05$, $^{**} = p \leq 0.01$; $^{***} = p \leq 0.001$; $^{****} = p \leq 0.0001$ (for full statistical results see S5 Table).

(Fig 1; full results are available at S5 Table). In particular, we observed a significant depletion of Denisovan and Neanderthal aSNPs from PolyComb repressed (ReprPC $p_{Deni} = 7.7 \cdot 10^{-65}$), bivalent enhancers, promoters and transcription start sites (EnhBiv $p_{Deni} = 5.0 \cdot 10^{-69}$, BivFlnk $p_{Deni} = 1.74 \cdot 10^{-20}$ and TssBiv $p_{Deni} = 4.95 \cdot 10^{-20}$, respectively). We also observed a Neanderthal-specific significant enrichment for heterochromatin (Het $p_{Nean} = 2.9 \cdot 10^{-18}$), whereas Denisovan aSNPs are significantly depleted from this state ($p_{Deni} = 4.6 \cdot 10^{-6}$). An exception to this trend was an excess of aSNPs from both archaic ancestries that fell within quiescent chromatin (Quies $p_{Deni} = 1.5 \cdot 10^{-15}$), as previously reported by [12]. This observation is not surprising given that quiescent chromatin is the predominant state in each epigenome [34], and this finding might indicate that a substantial fraction of aSNPs are functionally inert in most contexts, although silencers have also been found enriched within this state [49].

When considering active chromatin states, we noticed a significant depletion of Denisovan and Neanderthal ancestries from transcription start sites (TssA $p_{Deni} = 3.4 \cdot 10^{-4}$), promoters (TssAFlnk $p_{Deni} = 0.026$) and states marked by both promoter and enhancer epigenetic marks (TxFlnk $p_{Deni} = 0.0093$; Fig 1) and of Neanderthal aSNPs from enhancers (Enh, $p_{Nean} = 1.3 \cdot 10^{-8}$). On the other hand, we observed a significant enrichment for Denisovan and Neanderthal aSNPs within genic enhancers (EnhG $p_{Deni} = 1.6 \cdot 10^{-4}$) transcribed (Tx $p_{Deni} = 3.9 \cdot 10^{-30}$) and within zinc-finger protein-genes and repetitive sequence (ZNF/Rpts $p_{Deni} = 0.033$)

states, this latter particularly for Neanderthal variants (Fig 1). Finally, we found a significant enrichment of Denisovan aSNPs within weakly transcribed regions (TxWk $p_{Deni}$ = 2.3 · $10^{-10}$) (Fig 1). Importantly, we did not find any substantial differences in these patterns when considering introgressed SNPs regardless of allele frequency (S5 Fig and S6 Table).

To validate this annotation approach we repeated our analyses using a set of 11,422,574 Neanderthal variants identified within 75 individuals of West Eurasian ancestry [31], which we processed as above. We found similar patterns of enrichment across chromatin states in the Papuan and West Eurasian datasets, with Neanderthal variants being significantly enriched within Quies, Het, Tx and TxWk states (S6(A) Fig). Similarly to [12], we also observed a significant depletion of Neanderthal variants from Enh and EnhBiv, although we detected a significant enrichment within EnhG (S6(A) Fig); however, the three states were collapsed into a single category in [12] and fewer SNPs were associated with this state than with Enh. These results not only validate our SNP filtering criteria; they also suggest that, on a genome-wide scale, different instances of admixture with archaic hominins might broadly contribute to modern human populations primarily through gene regulatory change.

Finally, to confirm whether these signals were caused by SNPs occurring within highly pleiotropic or cell-specific functional elements, we counted the number of different cell types across which aSNP and naSNP -containing elements were annotated with any given chromatin state. This yielded an estimate of the potential pleiotropic activity of each functional element. We observed similar degrees of pleiotropy across all the three ancestries, with variants within Quies, TssA, Tx and TxWk exhibiting higher pleiotropic activities than those within TssAFlnk, TxFlnk, Enh and EnhG (S7 Fig). Given that 64,944 Denisovan, 41,667 Neanderthal and 108,282 non-archaic variants fell within chromatin states associated with transcribed regions and CREs in at least one cell type (Fig 1 and S5 Fig), these results indicate that a non-trivial fraction of introgressed variants might have functional consequences, particularly for gene regulatory processes occurring in a limited number of cell types.

## Denisovan and Neanderthal aSNPs might have different effects on gene regulatory processes across tissues

Given the observed enrichment of aSNPs within chromatin states associated with CREs, we sought to investigate the effects of Denisovan and Neanderthal aSNPs on gene regulation. Hence, we restricted our analyses to all common-to-high-frequency SNPs within TssA, TssaFlnk, TxFlnk, Enh and EnhG chromatin states, which resulted in three sets of 39,636, 24,653 and 67,098 Denisovan, Neanderthal and non-archaic variants. Next, to identify tissues containing an over-representation of Denisovan and/or Neanderthal variants, for each tissue we again computed the OR between the number of aSNPs and naSNPs (see Methods). Despite roughly similar numbers of aSNPs falling within CREs across all tissue types, patterns of enrichment vary substantially by tissue type (Fig 2; for full results see S7 Table). For instance, Denisovan and Neanderthal aSNPs are significantly depleted from iPSCs ($p_{Deni}$ = 1.2 · $10^{-7}$, $p_{Nean}$ = 9.6 · $10^{-5}$) and Neurospheres ($p_{Deni}$ = 0.02, $p_{Nean}$ = 0.0016), but significantly enriched within mesenchymal ($p_{Deni}$ = 1.7 · $10^{-7}$, $p_{Nean}$ = 3.3 · $10^{-5}$) and blood & immune T cells ($p_{Deni}$ = 3.5 · $10^{-15}$, $p_{Nean}$ = 3.3 · $10^{-6}$). In addition, enrichment patterns vary by aSNP ancestry, with Neanderthal-specific enrichment within smooth muscle ($p_{Nean}$ = 8.2 · $10^{-6}$) and Denisovan aSNPs significantly overrepresented within haematopoietic stem cells (HSCs) & B cells ($p_{Deni}$ = 5.0 · $10^{-4}$).

As above, repeating this analysis in Neanderthal aSNPs segregating within Europeans also identified significant enrichments of Neanderthal aSNPs within CREs active in mesenchymal and blood & T cells (S6(B) Fig). Interestingly, these findings fully recapitulate previously

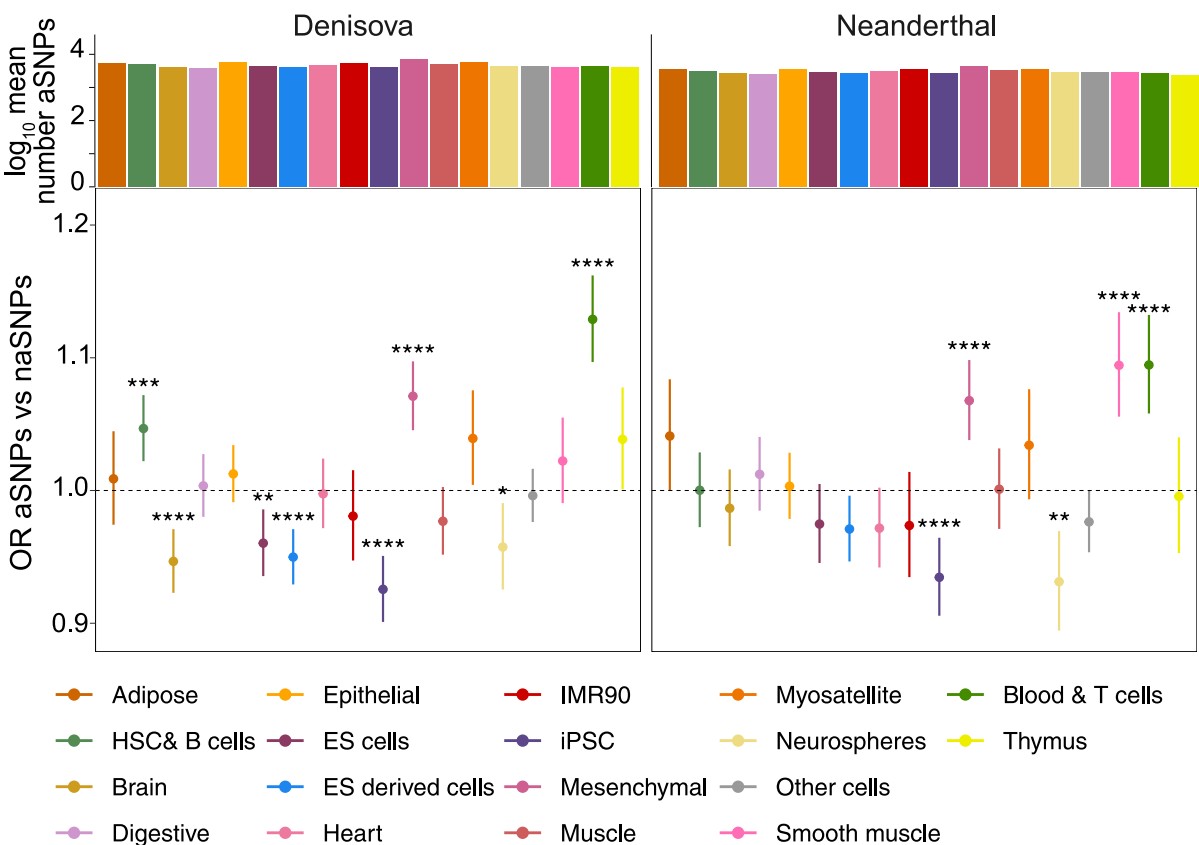

**Fig 2. Distribution of common-to-high-frequency aSNPs falling within CREs across 18 tissues.** Patterns of enrichment across the 18 different tissue types in Roadmap Epigenomics for the set of common-to-high-frequency aSNPs annotated within chromatin states associated with CREs, relatively to the matched background set of naSNPs. Histograms on top indicate the mean number of variants calculated across all cell types belonging to each tissue. Asterisks indicate BH-corrected Fisher's exact test p-values: * = $p \leq 0.05$, ** = $p \leq 0.01$; *** = $p \leq 0.001$; **** = $p \leq 0.0001$ (for full statistical results see S7 Table).

reported observations in Europeans [11, 12]. Thus, in line with previous studies [11, 22, 50–52], our observation of a strong enrichment of Denisovan aSNPs within CREs active in immune cell types would suggest a potential contribution of these archaic variants to regulatory processes happening within immune-related cells, and by extension, to immune-related traits.

As a final test, we intersected these variants with significant cis-eQTLs (qval < 0.05) from GTEx v8 [35]. We found 59 Denisovan, 117 Neanderthal and 388 non-archaic variants (0.15%, 0.47% and 0.58% of all common-to-high-frequency SNPs, respectively) were identified as potential cis-eQTLs in at least one tissue in GTEx. We expanded the analysis to the entire set of variants, regardless of their allele frequency, and found 132 Denisovan, 242 Neanderthal and 688 non-archaic variants (0.09%, 0.27% and 0.3%, respectively) overlapping with cis-eQTLs. Given the well-recognised bias towards individuals of European ancestry within the GTEx dataset, these findings both support the need to include under-represented populations into genetic surveys, and reaffirm that the majority of our introgression assignments are correct, and, as expected, that our variants are geographically restricted.

## Assessing the impact of aSNPs on transcription factor binding sites

We next sought to characterise whether aSNPs might alter gene regulation and the functional mechanisms by which this may occur. We thus examined the potential of aSNPs annotated

within CREs to disrupt known transcription factor binding sites (TFBS) by assessing their impact on 690 different position weight matrices (PWMs) drawn from the from HOCO-MOCO [40] and Jaspar [39] databases. After consolidating variants predicted to disrupt the same motif across the two databases (see Methods), we found that 16,048 Denisovan, 10,032 Neanderthal aSNPs and 28,370 non-archaic naSNPs (respectively 40.5%, 40.7% and 42.3% of the tested variants) were predicted to disrupt at least one TFBS. To avoid redundancy due to similarities between predicted motifs across closely related TFs, we took advantage of recent work by Vierstra *et al.* [41], which clusters HOCOCOMO and Jaspar PWMs into 286 distinct clusters on the basis of sequence similarity, and considered only those instances included in these clusters.

We then tested whether any of the motif clusters contained a significant excess of TFBS-disrupting aSNPs relative to naSNPs. Overall, we did not find any significant differences between the number of TFBS-disrupting archaic and non-archaic SNPs across the set of motif clusters (p = 0.26 and 0.14 for Denisovan and Neanderthal aSNPs, respectively). This suggests that, on a genome-wide scale, aSNPs are not disrupting any specific family of DNA motifs. To quantify the impact of aSNPs on the relevant PWM, for each of these TFBS-disrupting variants we then calculated a $\Delta$ PWM score, defined as the difference in the PWM score between the archaic and non-archaic alleles in case of aSNPs or between the derived and ancestral allele in case of naSNPs. While naSNPs are on the whole predicted to be significantly more disruptive than aSNPs, this difference is likely caused by the number of TFBS-disrupting variants reported in each category as $\Delta$ PWM scores remain highly comparable across ancestries (S8(B) Fig). Together, these results indicate that despite a lack of specific gene regulatory networks preferentially altered by aSNPs, a non-neglible fraction of our identified cis-regulatory variants might affect gene expression through altering the affinity between TFs and their cognate DNA sequences.

### High levels of population differentiation Denisovan aSNPs within CREs active in immune cells

Recent studies have shown the existence of genetic structure characterising the Indonesian archipelago, with Papuan—and thus, Denisovan—ancestry showing a marked west to east cline [6, 53]. We therefore investigated whether our subset of variants, particularly those within CREs active in immune-related cells, also showed similar levels of population differentiation across the region. To this extent, we defined aSNPs and naSNPs in haplotype data from 63 individuals from Western Indonesia, also part of [6]. From the total set of 14,117 Denisovan, 8,662 Neanderthal and 22,916 non-archaic variants annotated within CREs active in immune-related cells, we found that 4,732 Neanderthal and 9,682 non-archaic variants (42.2% and 54.6%, respectively) segregate within Neanderthal and non-archaic haplotypes in western Indonesians (Fig 3A). However, in keeping with the expectation of little to no Papuan genetic ancestry in Western Indonesia only 1,535 Denisovan variants (10.8%) in our Papuan sample are also segregating within Denisovan-like haplotypes in this additional data set (Fig 3A). Indeed, we found significantly higher $F_{st}$ values for the set aSNPs annotated within CREs, especially those of Denisovan ancestry ($p_{Wilcoxon} < 2.2 \cdot 10^{-16}$ and 0.0013, respectively for Denisovan and Neanderthal), compared to naSNPs (Fig 3B).

### Denisovan alleles are associated with gene expression differences in immune cells

Given the observed enrichment of aSNPs, especially of Denisovan ancestry, within the CREs of immune-related cells, we used GREAT [54] to link these to their nearest genes in an attempt

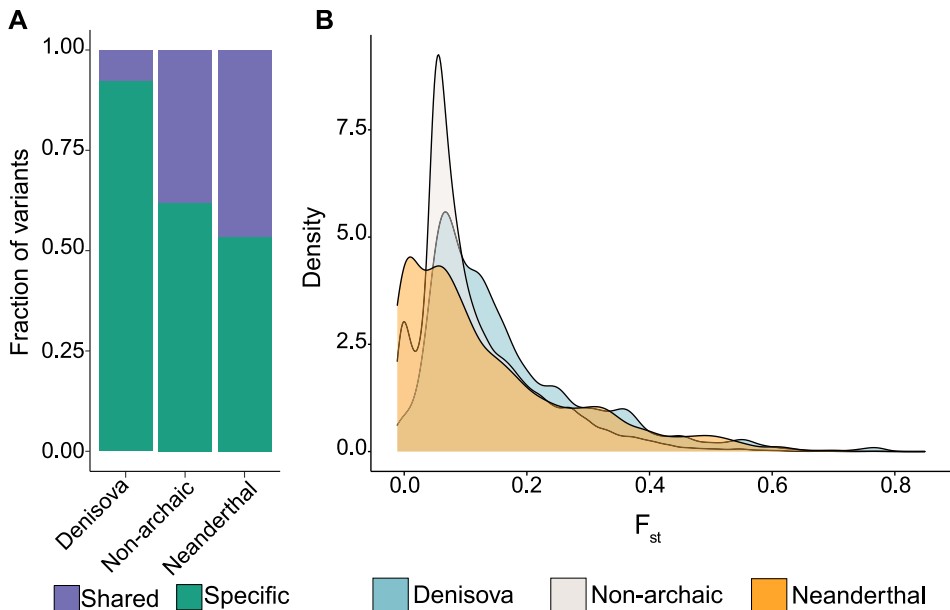

**Fig 3. Levels of population differentiation for the set of cis-regulatory SNPs between western Indonesian and New Guinean populations.** A) Proportion of variants shared across the Indonesian archipelago segregating within the corresponding haplotypes. B) Distribution of the $F_{st}$ values between western Indonesia and PNG for the three groups of variants.

to identify possible targets. There was limited overlap across ancestries in the sets of putative target genes. For instance, out of a total number of 9,504 putatively target genes only 303 (3.2%) are predicted to be targeted by all three ancestries (Fig 4A). Conversely, we found that 795 (8.3%) and 573 (6.0%) genes are uniquely targeted by Denisovan and Neanderthal variants, respectively (Fig 4A).

We next performed a Gene Ontology (GO) enrichment analysis on the set of genes associated with either archaic ancestry, using the set of genes associated with naSNPs as a bacground set. Denisovan aSNPs are associated with genes strongly involved in multiple immune-related processes (S8 Table). Genes targeted by Neanderthal variants are instead enriched for more general biological processes, albeit instances related to neutrophil/granulocyte migration and chemotaxis were observed for Neanderthal variants (S8 Table). We then computed the semantic similarity between the significantly enriched terms (GREAT reported BH-adjusted hypergeometric $p \leq 0.01$) using a method that incorporates both the locations of the terms in the GO graph as well as their relations with their ancestor terms [55], and found relatively low levels of semantic similarity between archaic and non-archaic enriched GO terms (Fig 4B), suggesting the set of variants might indeed affect the regulation of different biological processes within immune-related cells. Finally, for each ancestry we manually examined the set of genes associated with the significant GO terms. Denisovan cis-regulatory variants were predicted to regulate genes such as *TNFAIP3*, *OAS2* and *OAS3*, all of which have been repeatedly identified as harbouring archaic hominin contribution that impact immune responses to pathogens [51, 52, 56]. In particular, we found 20 Denisovan variants associated with *OAS2* and *OAS3*, eight of which (rs368816473, rs372433785, rs139804868, rs146859513, rs143462183, rs370655920, rs375463218 and rs372139279) were also predicted to significantly alter the affinity between TFs and their cognate DNA sequences by our analyses. Notably, in all cases the archaic allele segregated at frequencies between 0.2 and 0.4 in Papuans but was absent from Western

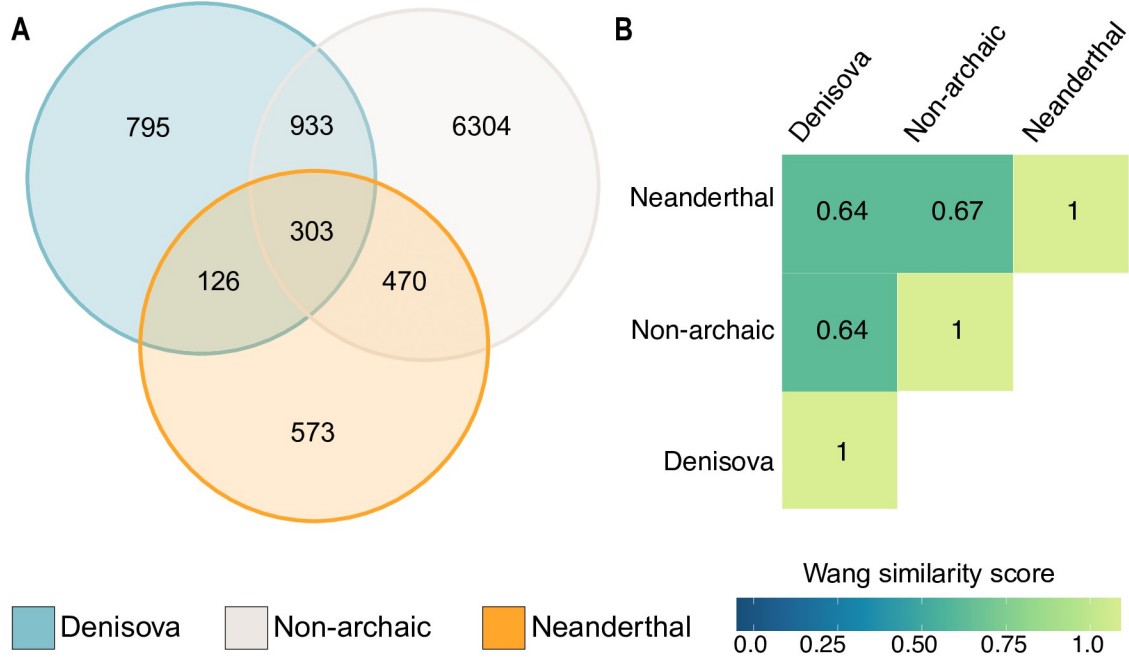

**Fig 4. Biological processes putatively affected by SNPs within immune-related CREs.** For each ancestry the figure shows **A)** the overlap between the putative sets of target genes for each ancestry; **B)** the semantic similarity score estimates between the set of significantly enriched GO terms.

Indonesia. Similarly, a comparison with 6 different non-human primate genomes indicated that for 7 of these 8 Denisovan aSNPs the introgressed allele is derived.

The *OAS* locus has been repeatedly found to harbour signals of both Neanderthal and Denisovan adaptive introgression, with archaic alleles predicted to alter gene regulation [52, 56]. Our previous work has shown that both *OAS2* and *OAS3* are differentially expressed in whole blood between the people of Mentawai, a small barrier island off the coast of Sumatra, in West Indonesia, and the Korowai, a genetically Papuan group living on the Indonesian side of New Guinea island (Fig 5B) [57]. To understand whether Denisovan introgression might contribute to these differences, we tested the regulatory activity of five of the eight Denisovan variants described above (rs372433785, rs139804868, rs146859513, rs143462183 and rs370655920), all of which were predicted to disrupt the binding sites of TFs active in human immune cells, using a plasmid reporter experiment (see Methods).

We tested all five alleles across two different lymphoblastoid cell lines (LCLs) established from Papuan donors. In all cases, we found the direction of effect to be consistent across biological and technical replicates. In particular, two Denisovan alleles (rs139804868:A>G and rs146859513:C>G) consistently showed significantly lower transcriptional rates compared to their non-archaic counterpart. rs139804868:A>G is predicted to disrupt a motif bound by *BHLHA15* and *TAL1*, which have been respectively found to be involved in immune B cell differentiation [58] and hematopoiesis [59]. rs146859513:C>G is predicted to disrupt a binding site for *NFKB2*, which is known to regulate the expression of cytokines and chemokines related genes [60] (Fig 5C). While confirming the validity of our approach, taken together, these findings point to a substantial contribution of Denisovan variants to immune-related processes in present-day Papuans, one chiefly mediated through the regulation of active immune responses mounted against pathogenic infections.

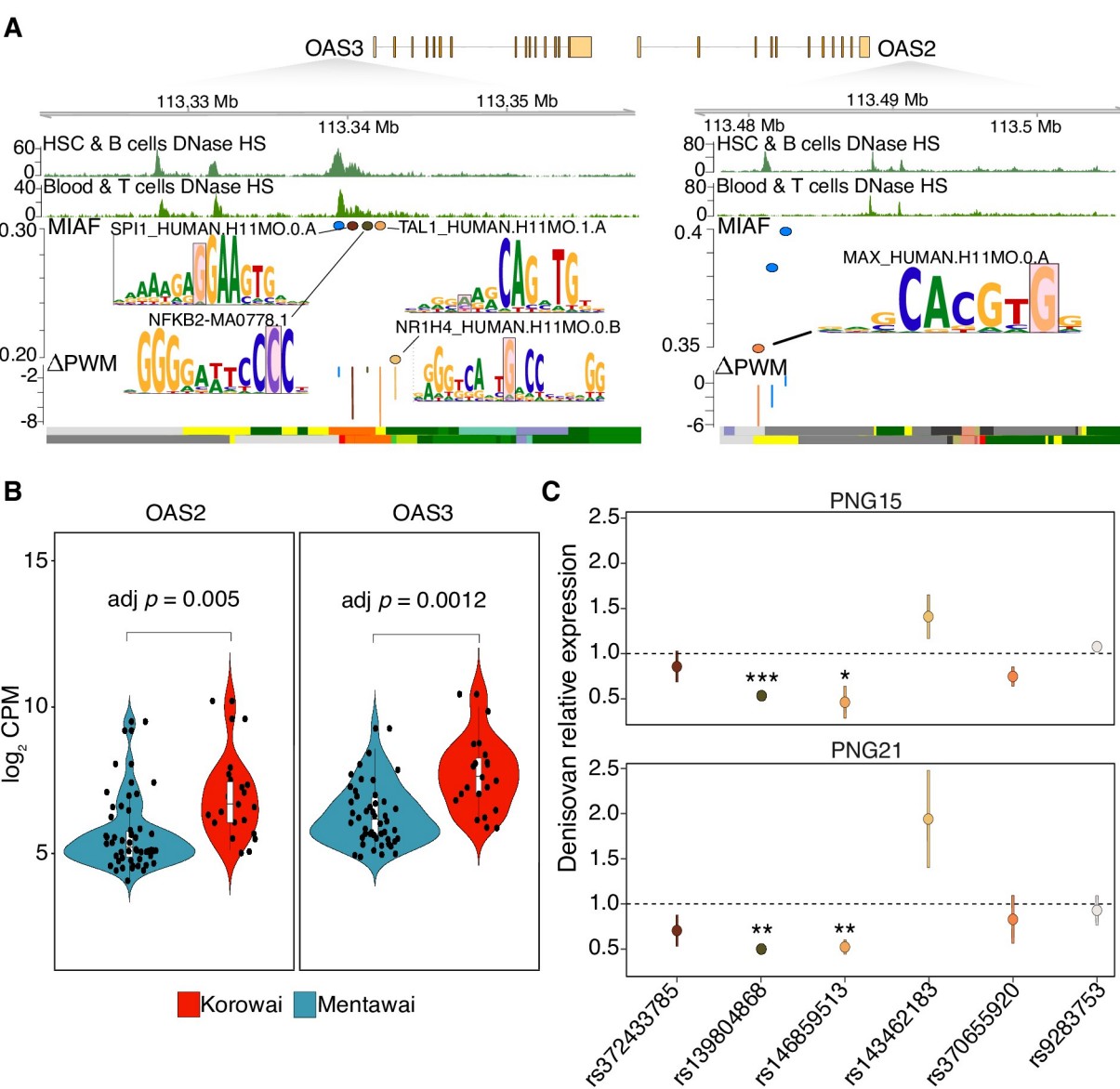

**Fig 5. Functional validation of the regulatory impact of Denisovan variants near *OAS2* and *OAS3*. A)** The genomic region encompassing the eight Denisovan variants associated with *OAS2* and *OAS3*. The top two tracks display patterns of DNase Hypersensitivity sites in Blood and immune T cells as well as in HSC and B cells [34]. Tested variants are shown along with their calculated Δ PWM. Bottom tracks display the chromatin state information for the same tissues; **B)** the distribution of the $\log_2$ RNA-seq counts per million in whole blood for *OAS2* and *OAS3* between the Korowai and the people of Mentawai, from [57]; **C)** the relative expression changes between Denisovan and non-archaic alleles, or between the alternative and the reference allele for positive control (rs9283753) [45], in two Papuan LCLs. Asterisks mark significant differences from 1, BH-corrected $p < 0.05$: *** = $0.0001 < p \leq 0.001$; ** = $0.001 < p \leq 0.01$; * = $0.01 < p \leq 0.05$.

## Discussion

There is significant interest in understanding the functional consequences of archaic introgression. Evidence indicates that both Denisovan and Neanderthal aSNPs, especially those within protein coding and conserved non-coding elements, are mostly deleterious and negatively selected in modern humans [14, 61]. Similar findings have been recently reported for highly pleiotropic enhancers, where aSNPs are depleted likely as a consequence of their potential to perturb gene expression across multiple tissues [12]. Nevertheless, out of the substantial

number of archaic variants still segregating within present-day populations, a large fraction falls within genomic regions that show strong evidence of functional activity across a variety of cell types. Indeed, studies conducted primarily on Neanderthal introgressed DNA have suggested a non-negligible contribution to gene expression variation in modern humans [10, 11, 22], with repeated examples of Neanderthal archaic variants falling within regulatory elements or the seed region of mature micro-RNAs predicted to affect transcriptional and post-transcriptional regulatory processes [11, 62].

In this study, we have taken advantage of a recently published dataset [6] to investigate the landscape of archaic introgression in individuals of Papuan genetic ancestry, the functional consequences of which remained poorly understood. This has allowed us to characterise the putative contribution of Denisovan DNA, which is known to account for up to 5% of the genome of present-day Papuans [4, 63], while also comparing it to that of Neanderthal DNA and non-archaic variants that arose following the Out of Africa event. We specifically analysed all of our variants across multiple cell types and functional chromatin states aiming to account for the strong dependency on the cells' chromatin landscapes [34] when assessing the potential activities of introgressed alleles. While our analyses focus on a small subset of introgressed variants identified by [6], our filtering criteria ensure that we only consider variants with a high likelihood of being truly introgressed. In addition, comparing Denisovan and Neanderthal signals within the same samples serves as an internal control for difference in linkage disequilibrium and background selection between ancestries, hence providing stronger support for any eventual Denisovan- and/or Neanderthal-specific contribution to phenotypic variation in contemporary Papuans.

First, we confirm previous reports that aSNPs mostly occur within non-coding sequences [9, 14, 64], with a large number falling within highly constitutive and/or functionally inert regions, which might be the result of weaker purifying selection acting on these elements. aSNPs are significantly depleted from other inactive chromatin states but over-represented within states putatively containing elements involved in gene regulatory processes. Applying our approach to Neanderthal aSNPs segregating within present-day West Eurasians [31] replicates most of our findings, as well as previously reported signals including a depletion of Neanderthal variants within enhancer-like elements [12]. Previous studies have shown that, while polygenic risk scores [65] and eQTLs [66] have limited cross-population portability, biochemically active functional regulatory elements tend to be under evolutionary constraint within the human lineage [67, 68]. Thus, while our approach is contingent on functional data generated in a population that is not of Papuan ancestry, our focus on functional analyses are likely to be more robust to differences across populations in linkage disequilibrium. Nevertheless, our set of SNPs, and our conclusions, represent a first step that will benefit from functional validation at scale.

Second, we find notable differences between the two archaic ancestries in their patterns of introgression. Such differences are likely consequence of the tissue-specificity nature of CREs, and of enhancers in particular [69]. Indeed we report that variants annotated in these CRE-associated states tend to lie within elements active in a restricted number of cell types, suggesting that the effects of archaic introgression might be highly cell-type-specific. In addition, admixture with Neanderthals might have similar consequences even across distantly related human populations, with Neanderthal aSNPs strongly enriched within elements active within blood & immune T cells and mesenchymal cells both in Papuans and Europeans, replicating previously reported observations [11, 12]. While we report a similar enrichment for Denisovan alleles within these tissues, we further note a Denisovan aSNPs-specific significant enrichment with CREs active within HSC & B cells, suggesting a more pervasive contribution of Denisovan introgression to gene regulatory processes happening within immune-related cells. In line

with this, we further show that Denisovan and Neanderthal variants annotated within immune-related CREs might target different sets of genes with limited overlap in the biological processes they are involved in. Indeed, only genes predicted to be regulated by Denisovan aSNPs are strongly involved in active immune responses.

Third, we have characterised the impact of introgressed variants on transcription factor binding sites. A substantial proportion of aSNPs within CREs are predicted to modify the interactions between TFs and their cognate binding sites, although we do not detect an overall higher impact of aSNPs relative to naSNPs. Our data suggest that, despite aSNPs possibly disrupting individual DNA motifs, at a genome-wide level admixture with archaic hominins is unlikely to have resulted in large rewiring of transcription factor regulatory networks. While this might contrast the findings of previous studies [70], in our study we focused on families of DNA motifs (as clustered by [41]) rather than consider enrichment within individual PWMs, an approach that we believe is better suited to controlling for the typical redundancy of PWMs. This, however does not exclude that aSNPs might not disrupt specific DNA motifs. Indeed, our approach also identifies Neanderthal aSNPs disrupting the PWM of TFs such as *CREB1*, *USF1*, *MYB* and *STAT6*, similarly to what has been reported by [70]. We also note that there is a significant (hypergeometric $p < 2.2 \cdot 10^{-16}$) overlap between genes targeted by N-eQTLs in [70] and our own set of possible target genes (S9 Fig).

Finally, we performed a plasmid reporter assay experiment to quantify the molecular impact of Denisovan TFBS-disrupting variants annotated within CREs active in immune-related cells and predicted to affect the expression level of *OAS2* and *OAS3*. These genes belong to a family of pattern-recognition receptors involved in innate immune responses against both RNA and DNA viruses, with *OAS3* considered to be essential in reducing viral titer during Chikungunya, Sindbis, influenza or vaccinia viral infections [71]. At least two previous studies have shown the presence of both Neanderthal and Denisovan archaic haplotypes segregating at this locus respectively within European [52] and Papuan [56] individuals. Sams *et al.* found two variants (rs10774671, rs1557866) within these Neanderthal haplotypes which are respectively associated with the codification of different *OAS1* splicing isoforms and with a reduction in *OAS3* expression levels, the latter only upon viral infection [52]. Our previous work has found that both *OAS2* and *OAS3* are differentially expressed between western Indonesians and Papuans [57]. Here we report a set of eight SNPs (rs368816473, rs372433785, rs139804868, rs146859513, rs143462183, rs370655920, rs375463218 and rs372139279), located roughly 41 kb upstream of *OAS2* and *OAS3*, all of which are predicted to strongly alter the ability of different transcription factors, including *IRF4*, *NFKB2* and *TAL1*, to bind to their underlying DNA motifs. In all cases, the reference allele is fixed within western Indonesian populations, whereas the archaic alleles segregate at frequencies between 0.2 and 0.4 in Papuans. We show that at least 5 of these variants lie within sequences that can regulate expression in reporter gene plasmid assays, and that in two of these variants (rs139804868 and rs146859513) the Denisovan allele is associated with significantly lower transcriptional activity compared to the non-archaic allele in immune cells from two different Papuan donors, again suggesting that these SNPs are of biological importance.

Recent work by [72] has shown that variants which arose in the human lineage following its split from the Neanderthal-Denisovan common ancestor, and which have reached (near) fixation in modern humans since then can exhibit significant differences in regulatory activity relative to the ancestral state. Similarly, Jagoda *et al.* has shown large impact on gene expression associated with the presence of Neanderthal aSNPs in vitro [73]. Our results suggest that Denisovan alleles segregating within modern human populations are also likely to actively participate in gene regulatory processes, especially those that take place within immune-related cells. This agrees with recent findings from a study that analysed the genome of present-day people

of Taiwan, the Philippines, the Solomon Islands and Vanuatu [50]. While further experimental validation of our observations is necessary in order to characterise the genome-wide impact of archaic introgression, the results presented in this study argue for a potential contribution of Denisovan variants to immune-related phenotypes amongst early modern humans in the region, potentially favouring adaptation to the local environment [15, 22].

## Supporting information

**S1 Table. Table reporting the number of SNPs retained after each filtering step.**
(XLSX)

**S2 Table. File reporting the genomic location of all Denisovan variants used in this study.**
(TXT)

**S3 Table. File reporting the genomic location of all Neanderthal variants used in this study.**
(TXT)

**S4 Table. File reporting the genomic location of all non-archaic variants used in this study.**
(TXT)

**S5 Table. Table reporting the enrichment results for the set of common-to-high-frequency archaic variants annotated within each chromatin state.**
(XLSX)

**S6 Table. Table reporting the enrichment results for the entire set of archaic variants annotated within each chromatin state.**
(XLSX)

**S7 Table. Table reporting the enrichment results across tissues for the set of common-to-high-frequency archaic variants annotated within chromatin states associated to CREs.**
(XLSX)

**S8 Table. Table reporting the GO enrichment results for the set of genes targeted by Denisovan, Neanderthal and non-archaic variants.**
(XLSX)

**S9 Table. Table reporting the primer and oligo sequences used in the *in-vitro* validation.**
(XLSX)

**S1 Fig. aSNPs distribution across haplotypes.** Histogram showing the distribution of the Denisovan and Neanderthal allele frequency differences between archaic and non-archaic haplotypes. Negative values indicate putative archaic variants segregating at higher frequencies within non-archaic haplotypes.
(PDF)

**S2 Fig. Overview of site frequency spectrum (SFS).** Histogram showing the SFS for Denisovan, Neanderthal and non-archaic variants **A)** before and **B)** after the variant filtering steps. **C)** Histogram showing the SFS for aSNPs (Neanderthal + Denisovan) and the matched background set of naSNPs.
(PDF)

**S3 Fig. Estimates of the levels of background selection.** Plots showing the distribution of the B-statistic values for the genomic regions containing the refined set of **A)** all Denisovan, Neanderthal and non-archaic variants or **B)** only the common-to-high-frequency SNPs. Lower

values indicate higher evolutionary constrains. For visual comparison, the distribution of the B-statistic values for all human protein coding sequences is also shown. Reported p-values are calculated from Mann-Whitney U test.
(PDF)

**S4 Fig. Genome-wide functional annotation of aSNPs and naSNPs.** Histogram showing the proportion of **A)** all aSNPs and naSNPs and **B)** only common-to-high-frequency variants annotated across multiple genomic elements as reported by annotatr [64]. Percentages are relative to the SNPs annotated across the categories shown. Total numbers of variant annotated: **A)** 72,761 Denisovan, 44,135 Neanderthal and 114,518 non-archaic; **B)** 42,334 Denisovan, 27,400 Neanderthal and 68,310 non-archaic.
(PDF)

**S5 Fig. Distribution of all aSNPs across 15 chromatin states.** Figure showing the patterns of enrichment across the 15 chromatin states for the entire set of Denisovan and Neanderthal aSNPs relatively to the matched background set of naSNPs. Histograms on top indicate the mean number of variants within each chromatin state calculated across all 111 cell types. Asterisks indicate BH-corrected Fisher's exact test p-values < 0.05, i.e., **** = $p \leq 0.0001$; *** = $0.0001 < p \leq 0.001$; ** = $0.001 < p \leq 0.01$; ** = $0.01 < p \leq 0.05$ (for full statistical results see S6 Table).
(PDF)

**S6 Fig. Distribution of Neanderthal aSNPs identified within EUR across chromatin states and tissues.** Figure showing the patterns of enrichment across **A)** the 15 chromatin states for the entire set of Neanderthal aSNPs **B)** the 18 different tissues for the set of cis-regulatory Neanderthal variants segregating at common-to-high-frequencies in EUR. OR are computed relatively to the matched background set of naSNPs. Histograms on top indicate the mean number of variants annotated within each chromatin state (A) or within each tissue (B). The mean is respectively calculated across all 111 cell types and across the cell types belonging to each tissue. Asterisks indicate BH-corrected Fisher's exact test p-values < 0.05, i.e., **** = $p \leq 0.0001$; *** = $0.0001 < p \leq 0.001$; ** = $0.001 < p \leq 0.01$; ** = $0.01 < p \leq 0.05$.
(PDF)

**S7 Fig. Pleiotropic activities of chromatin state-associated elements carrying SNPs.** Cumulative proportion of the pleiotropic activity across 111 cell types of each chromatin state-associated element carrying SNPs.
(PDF)

**S8 Fig. SNPs impact on TFBSs.** Distribution of the Δ PWM scores for the set of aSNPs and naSNPs. P-values are returned from Wilcoxon test.
(PDF)

**S9 Fig. Overlap between target genes identified by Findley et al and this study.**
(PDF)

## Acknowledgments

We would like to thank all members of the Gallego Romero and McCarthy groups for their helpful discussions and comments on the manuscript. We also wish to acknowledge all of the study participants who generously consented to genome sequencing in the original study, and the leadership of Herawati Sudoyo and Chelzie Crenna Darusalam (Eijkman Institute for Molecular Biology) in generating these datasets.

## Author Contributions

**Conceptualization:** Davide M. Vespasiani, Irene Gallego Romero.

**Data curation:** Guy S. Jacobs.

**Formal analysis:** Davide M. Vespasiani, Guy S. Jacobs.

**Funding acquisition:** François-Xavier Ricaut, Irene Gallego Romero.

**Investigation:** Davide M. Vespasiani, Laura E. Cook, Nicolas Brucato.

**Methodology:** Davide M. Vespasiani, Laura E. Cook, Nicolas Brucato.

**Project administration:** Nicolas Brucato, Irene Gallego Romero.

**Resources:** Guy S. Jacobs, Nicolas Brucato, Matthew Leavesley, Christopher Kinipi, François-Xavier Ricaut, Murray P. Cox.

**Supervision:** Murray P. Cox, Irene Gallego Romero.

**Writing – original draft:** Davide M. Vespasiani, Irene Gallego Romero.

**Writing – review & editing:** Davide M. Vespasiani, Guy S. Jacobs, Laura E. Cook, Nicolas Brucato, Matthew Leavesley, Christopher Kinipi, François-Xavier Ricaut, Murray P. Cox, Irene Gallego Romero.

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
