## [Decision Letter · Decision Letter 0]

29 Mar 2022

Dear Dr Gallego Romero,

Thank you very much for submitting your Research Article entitled 'Denisovan introgression has shaped the immune system of present-day Papuans' to PLOS Genetics.

The manuscript was fully evaluated at the editorial level and by independent peer reviewers. The reviewers appreciated the attention to an important problem, but raised some substantial concerns about the current manuscript. Based on the reviews, we will not be able to accept this version of the manuscript, but we would be willing to review a much-revised version. We cannot, of course, promise publication at that time.

If you decide to revise the manuscript for further consideration at PLOS Genetics, please aim to resubmit within the next 60 days, unless it will take extra time to address the concerns of the reviewers, in which case we would appreciate an expected resubmission date by email to plosgenetics@plos.org.

[LINK]

We are sorry that we cannot be more positive about your manuscript at this stage. Please do not hesitate to contact us if you have any concerns or questions.

Yours sincerely,

Anna Di Rienzo

Associate Editor

PLOS Genetics

Hua Tang

Section Editor: Human Variation

PLOS Genetics

Reviewer's Responses to Questions

**Comments to the Authors:**

Reviewer #1: In this paper, Vespasiani et al investigate the functional genetic legacy of

Neandertal and Denisovan ancestry on present-day Papuans. Overall, I had some

difficulty assessing the functional genetic analyses presented in this paper,

largely because they fall outside my area of expertise, but also because the

paper does not provide a lot of background, and in general does not justify

specific filters and analyses. Thus, as currently written, following most analyses

requires specific knowledge of the used packages, which I largely lack.

Thus, my concerns are largely about the technical validity and clarity of the

manuscript:

1. Predicting phenotypes across human populations

- In the introduction, the authors point out that it remains difficult to work in non-Europeans

because most reference data sets are produced in European-ancestry data sets.

One well-known drawback, for example, is that genomic predictions of quantitative traits

(even simple traits such as height) from one population to another have very low accuracy. Here,

the authors use functional annotations made in Non-Papuans to infer the effect

of archaic ancestry in Papuans. How consistent and transferable are these functional genetic

categories between populations?

2. The filtering of SNPs is at times poorly described and justified.

The authors restrict their analyses to a very small percentage of sites (<5%), following a set of stringent

filtering steps. While there is some justification for these steps, my worry is

that the concentration on such a small proportion of the data does introduce

some biases. At the very least, it would be nice to know which step removes how

many sites, and how basic properties (such as the SFS of aSNP/naSNP) change

with each filtering step. For example, does restricting analyses to aSNP at frequency

> 0.25 (but not having a corresponding filter for naSNP) impact the results?

I also have a hard time following the filters applied:

- The definitions of allele frequencies are either imprecise or wrong. For example,

(p21) why is the allele frequency not calculated as (number of variant alleles) / (total

number of alleles)? Why is there a normalization by ancestry for allele

frequencies, but not DAF and MIAF? Why do you not distinguish between

homozygotes and heterozygotes?

As defined, allele frequencies are on Papuan individuals, so a different definition must exist for sub-saharan Africans (next

paragraph)

- the manuscript is also vague about what filtering steps were done for haplotypes, and what steps were done on

SNPs, and how LD was incorporated?

3. Is allele frequency a confounder?

Many functional categories targeted by selection are enriched at certain allele

frequencies, e.g. we would expect deleterious alleles to be more rare.

Due to the rather tight timing of introgression, introgressed alleles are

typically much more constrained in their age (and hence allele frequency), i.e.

there are no very old or very young introgressed alleles. Can you show whether

there is any effect of archaic ancestry beyond allele frequency differences? If

the shown results are just a byproduct of differences in allele frequency, this

would be important to mention, in particular since aSNP and naSNP are

differentially filtered. One possibility I could imagine this could work is if the

allele frequencies were matched between naSNP and aSNP.

4. No validation of methods.

There have been several papers that have looked at the functional impact of

Neanderthal ancestry in present-day Europeans (most of them cited in the paper).

As the authors have their previously called haplotypes already, I understand why

they chose to use a somewhat different set of filters and analyses compared to

previous analyses, but one way that would raise my confidence in the presented

pipeline were contrasted with previous approaches. At the very least, the

results on Neanderthals should be contrasted with previous approaches in the

discussion, but many of my concerns regarding adequacy of the method could be

addressed by demonstrating that the presented pipeline works well and can

reproduce previous results on e.g. Europeans.

Reviewer #2: Vespasiani et al. report their findings on the functional make up of genomic regions of Neanderthal and Denisovan origin in present day Papuans. As the authors mention, it is very important that previously neglected groups can also benefit from genomic studies that illuminate their past evolutions and the potential consequences for health, especially immune diseases in that case. In this case, the authors present evidence that the expression regulation of immune genes in present day Papuans has been shaped by adaptive introgression from especially Denisovan segments of DNA. Given past literature it is a very reasonable claim to make. That said, the authors will greatly strengthen their claim by adding a number of crucial controls that are currently missing form their analyses.

My main concern about the manuscript is that important controls, in the spirit of the controls implemented for example by Francesca Luca in Plos Genetics “A signature of Neanderthal introgression on molecular mechanisms of environmental responses”. It is now well known that archaic introgression is determined at the local genomic level by a combination of the local level of recombination and the local level of deleterious variation/background selection. So any over-represented immune function found by the authors could be due to this function having on average higher recombination and/or less background selection on average, rather than higher rates of adaptive introgression. This is currently a major weakness of the manuscript. It is unclear whether what the authors found is due to less background selection/higher recombination or due to genuine adaptive introgression having shaped immune regulation. The former is much more trivial and unsurprising than the latter. The authors should check what Luca et al. did to control whether or not recombination and background selection need to be controlled for. If so, the authors need to recalculate the odds ratios while matching levels of recombination (from the deCode 2019 map that is not biased by the effect of selection on LD) and levels of background selection as measured by McVicker’s B. With this done the authors will be able to make a stronger claim for pathogens in the Papuan peoples’environments having shaped their immune genes through adaptive introgression.

Another important control that is missing and is potentially important given the results with the OAS genes and with immune over-representation overall is to control for the potential confounding effect of gene clustering. Just as the OAS genes, many immune genes are clustered together in the human genome. Therefore, it is difficult to know from the current results, whether some odds ratios are higher due to a genuine enrichment, or are spurious and due to the retained loci being clustered together. This is a pretty serious missing thing in the manuscript in its current form. The authors need to control for this, for example by counting important variants next to each other as only one variant. If they don’t do it, it could just be that the highlighted immune functions just happen to be functions that have high clustering. We can know this just from looking at where genes are in the genome, and that would weaken the claims made by the authors.

Minor concerns:

P3: traits of possible evolutionary advantage: misses many important citations. A few that come to mind: Gouy & Excoffier MBE 2020, Enard & Petrov Cell 2018.

Figure 1 is unreadable.

P7. Evolutionary importance and frequency higher than 0.2. It is very unclear and confusing what this means. It is insufficient to claim that the variants may be under positive selection. There are many more tests to evaluate such evidence, and the choice of the threshold is not justified.

Reviewer #3: The manuscript by Vespasiani and colleagues investigates the functional significance of introgressed Denisovan and Neanderthal variants in the context of transcription factor binding and gene regulation. While a few studies have analyzed Neanderthal introgressed variants and their role in gene regulation and complex trait variation in modern humans, much less is known about the potential function of non-coding variants that were introgressed from the Denisovan genomes. This is partly due to the limited availability of genomes and functional genomic data from populations where Denisovan introgression is present. This study provides a useful annotation of Denisovan introgressed variants with experimental validation of a study case for the OAS genes. The study is interesting and aligns with similar manuscripts that focus on Neanderthal introgression only, thus contributing to solve the puzzle of archaic introgression function in the human genome. Major and minor comments are listed below:

Major comments:

1. All variants that couldn’t be unambiguously attribute to only Neanderthal or Denisovan origin were excluded from the analysis. These could represent archaic introgressed variants that were shared between Neanderthals and Denisovans. It would be more informative to consider them as a third group of variants to investigate and include in the analysis, alongside with the Denisovan-specific, Neanderthal-specific, and Modern Human variants. Which TFBS do they regulate, are they tissue-specific? Are they enriched in immune response regulatory regions? Considering these variants would also help understand and discuss any difference between some of the results presented here and previous studies of Neanderthal introgressed variants.

2. The manuscript correctly mentions that the genomic distribution of introgressed variants relative to functional elements could be influenced by purifying selection. The manuscript should exclude that the observed genomic distribution of introgressed variants is due to background selection, for example by comparing the B-statistic values (McVicker et al, 2009).

3. Intersection with eQTLs: Were only lead/top eQTLs considered for this analysis? The overlap seems very limited, which is surprising when compared with previous studies that analyzed Neanderthal eQTLs only. Given that the GTEx eQTLs are not fine mapped, it would make sense to report other significant eQTLs that overlap with introgressed variants beyond the lead SNP.

4. How do the results on TFBS disrupting variants compare with previous studies of Neanderthal introgressed variants (e.g. Silvert et al 2019, Findley et al, 2021)? Please discuss the results in the context of previous studies.

5. What is the rationale behind using allele frequency correlation rather than FST to study the geographic distribution of archaic and modern human variants between the two Papuan and West-Indonesian populations? Is the FST distribution shifted for Denisovan variants compared to modern and Neanderthal variants? In general, there seems to be very little correlation in the plots despite the high R values reported.

Minor comments:

1. Please improve the quality of figure 1.

2. Please include a file (or link to a file in a publicly accessible location) with the rsID and genomic coordinates for the Denisovan and Neanderthal introgressed variants used in the study.

**Have all data underlying the figures and results presented in the manuscript been provided?**

Reviewer #1: Yes

Reviewer #2: Yes

Reviewer #3: **No: **A file (or link to a file in a publicly accessible location) with the rsID and genomic coordinates for the Denisovan and Neanderthal introgressed variants used in the study should be provided.

PLOS authors have the option to publish the peer review history of their article (what does this mean?). If published, this will include your full peer review and any attached files.

Reviewer #1: **Yes: **Benjamin Peter

Reviewer #2: No

Reviewer #3: No

---

## [Decision Letter · Decision Letter 1]

27 Sep 2022

Dear Dr Gallego Romero,

Thank you very much for submitting your Research Article entitled 'Denisovan introgression has shaped the immune system of present-day Papuans' to PLOS Genetics.

The manuscript was fully evaluated at the editorial level and by independent peer reviewers. The reviewers appreciated the attention to an important topic but identified some concerns that we ask you address in a revised manuscript.

We therefore ask you to modify the manuscript according to the review recommendations. Your revisions should address the specific points made by each reviewer.

[LINK]

Yours sincerely,

Anna Di Rienzo

Academic Editor

PLOS Genetics

Hua Tang

Section Editor

PLOS Genetics

Reviewer's Responses to Questions

**Comments to the Authors:**

Reviewer #2: The authors have carefully addressed my concerns.

Reviewer #3: This version of the manuscript by Vespasiani and colleagues presents extensive revisions to address all major comments by the three reviewers.

A couple of minor points remain unaddressed, but these should be pretty straight forward to take care of:

1. Point #3 by reviewer 3 (major comments). Please clarify in the methods section whether any filter/significance threshold was applied on the GTEx eQTL dataset used. Were multiple SNPs per gene considered? The number of archaic alleles annotated as eQTLs has increased in this revised version, but it is unclear why.

2. Point #2 by reviewer 3 (minor comment). Please make sure to include a file (or link to a file in a public repository) with the rs IDs and genomic coordinates for the Denisovan and Neanderthal introgressed variants used in the study, which is not limited to the ones annotated as putatively functional but includes all variants considered. In the current version, the two tables indicated in the response to reviewers’ comments contain the following: “Supplementary Table 5 Table reporting the GO enrichment results for the set of genes targeted by Denisovan, Neanderthal and non-archaic variants.

Supplementary Table 6 Table reporting the primer and oligo sequences used in the in-vitro validation.”

**Have all data underlying the figures and results presented in the manuscript been provided?**

Reviewer #2: Yes

Reviewer #3: **No: **A file (or link to a file in a public repository) with the rs IDs and genomic coordinates for the Denisovan and Neanderthal introgressed variants used in the study should be included. This file should not be limited to the variants annotated as putatively functional but should include all variants considered.

PLOS authors have the option to publish the peer review history of their article (what does this mean?). If published, this will include your full peer review and any attached files.

Reviewer #2: No

Reviewer #3: No

---

## [Editor Report · Decision Letter 2]

10 Oct 2022

Dear Dr Gallego Romero,

We are pleased to inform you that your manuscript entitled "Denisovan introgression has shaped the immune system of present-day Papuans" has been editorially accepted for publication in PLOS Genetics. Congratulations!

Yours sincerely,

Anna Di Rienzo

Academic Editor

PLOS Genetics

Hua Tang

Section Editor

PLOS Genetics

Comments from the reviewers (if applicable):

**Data Deposition**

http://datadryad.org/submit?journalID=pgenetics&manu=PGENETICS-D-22-00130R2

**Press Queries**

---

## [Editor Report · Acceptance letter]

18 Nov 2022

PGENETICS-D-22-00130R2 

Denisovan introgression has shaped the immune system of present-day Papuans 

Dear Dr Gallego Romero, 

We are pleased to inform you that your manuscript entitled "Denisovan introgression has shaped the immune system of present-day Papuans" has been formally accepted for publication in PLOS Genetics! Your manuscript is now with our production department and you will be notified of the publication date in due course.

With kind regards,

Anita Estes

PLOS Genetics

On behalf of:
